# Identifying Groundwater and River Water Interconnections Using Hydrochemistry, Stable Isotopes, and Statistical Methods in Hanumante River, Kathmandu Valley, Central Nepal

**Ramita Bajracharya [1], Takashi Nakamura [2,*], Subesh Ghimire [1], Bijay Man Shakya [2] and Naresh Kazi Tamrakar [1]**

[1] Central Department of Geology, Tribhuvan University, Kirtipur, Kathmandu 44613, Nepal; bajrarami@yahoo.com (R.B.); shghimire2001@gmail.com (S.G.); nktam555@gmail.com (N.K.T.)

[2] Interdisciplinary Centre for River Basin Environment (ICRE), University of Yamanashi, 4-4-37 Takeda, Kofu, Yamanashi 400-8511, Japan; inform.bj76@gmail.com

* Correspondence: tnakamura@yamanashi.ac.jp

**Abstract:** Interconnection between river water and groundwater plays an important role in maintaining water quantity and quality in hydrological systems. Furthermore, the exact interconnection is often difficult to observe and measure. This study attempts to explain river and shallow groundwater interconnection in urbanized areas of the Kathmandu Valley, Nepal. Isotopic ($\delta D$ and $\delta^{18}O$) and chemical analyses were performed on river and groundwater samples, and the results were analyzed using statistical methods to identify areas of interconnection between river water and groundwater. Higher concentrations and positive strong correlations of $Na^+$ with $K^+$, $NH_4^+$-N, $Cl^-$, $HCO_3^-$, and $PO_4^-$-P, and a change of water type from $Ca$-$HCO_3$ during the wet season to $Na$-$K$-$HCO_3$ during the dry season indicate higher contamination in river water during the dry season. Hierarchical cluster analysis was used in grouping water samples into clusters on the basis of isotopic and chemical ($Na+$ and $Cl^-$) composition. Grouping of river and groundwater samples in one–one clusters from wet and dry seasons shows the presence of interconnection, indicating the contribution of river water in recharging shallow groundwater. These results imply that shallow groundwater found near rivers is chemically contaminated by polluted river water through bank infiltration, in both wet and dry seasons.

**Keywords:** stable isotopes; chemical ions; hierarchical cluster analysis; groundwater-river water interconnection; Hanumante River

## 1. Introduction

Interconnection of river water and groundwater is a process of exchange between waters located on the river channel with those in the rocks/sediments under the surface. The exchange rate of water is controlled by hydraulic conductivities of the river channel and aquifer sediments; the relative stage of the river channel and nearby groundwater level; and geometry of the river channel within the alluvial plain [1,2]. The presence of a clogging layer on an aquifer and a riverbed or bank can decrease or stop the water flow exchange [3]. Interconnection of river water and groundwater also depends on the distance from a river channel, the geological conditions, and climatic factors [4]. Understanding of groundwater and surface water interconnection is very important to develop effective water resource management and policy as it can change the water quality and quantity of both water systems [5,6].

Studies related to the interconnection between river water and groundwater has increased in most developing countries during the last few decades. Utmost studies have been carried out to the assess areas of interconnection and the presence of exchange flow between river and groundwater on a regional as well as local scale [7–9]. Essentially, two types of exchange flow conditions are involved in the river and groundwater interaction: (1) the influent condition and (2) the effluent condition. Based on the condition of exchange flow, several studies specify that base flow in the river during the dry season is the result of effluent flow from shallow groundwater [5,10–13]. Furthermore, the condition of the exchange process can be affected by anthropogenic activities that alter the exchange processes, reduce connectivity, and lead to chemical or biological contaminations [2,14]. Increased sewage load into the rivers running through the urbanized cities can transfer toxic contamination to surrounding shallow groundwater in influent reaches. The decline of water table in a nearby shallow aquifer due to over-extraction can increase groundwater recharge from polluted river water [15,16]. The anthropogenic activities driven by increased urbanization and population growth affect river water quality, which adversely reflects on nearby groundwater quality.

Previous research conducted in the Kathmandu Valley reported the presence of an interconnection between river water and groundwater, showing a recharge of the groundwater by river water [17–19]. Additionally, a number of previous studies on river water quality have also reported heavy contamination of the downstream section of major rivers [20–23], inducing the occurrence of various water-borne diseases such as diarrhea, cholera, and dysentery among the people of riverside areas [24].

Methods such as heat tracer, solute tracer, direct measurement of water flux, and environmental tracer methods including isotope and geochemistry have been used to determine the interconnections of groundwater and surface water [25–30]. Similarly, numerical modeling, geophysical methods, and statistical methods have also been used to describe interaction processes [31–34]. Basically, water flux measurement mostly used a direct method to get exchange flow conditions involved in interacting processes. The recharging source for groundwater or river water is dependent on the direction of exchange flow [35]. Further, stable isotopes of hydrogen and oxygen along with Na and Cl ions have been widely used to determine river and groundwater interconnections. Similarity in isotopic values and chemical ions between nearby groundwater and river water indicate the presence of river and groundwater interconnection [13,36]. Additionally, hierarchical cluster analysis (HCA), one of the multivariate methods, has been widely used to determine the interaction of river water and groundwater at the absence of a water flux dataset [37]. HCA can be used to analyze regional-scale as well as local small datasets. Several previous studies used HCA as a major statistical method and suggested similarities in chemical and isotopic compositions between river water and their nearby groundwater representing the presence of river and groundwater interaction [9,34,37].

The present study will focus on the application of stable isotope values, chemical compositions of the river and groundwater samples, followed by statistical analysis for one of the contaminated rivers in urban areas of the Kathmandu valley. Thus, identifying the occurrence of the spatial and seasonal interconnectivity and possible contamination load from the river to river periphery groundwater are major goals of this study.

## 2. Materials and Methods

### 2.1. Study Area

The Hanumante River is a centripetal river and is one of the most polluted tributaries of the Bagmati River in the Kathmandu Valley, Central Nepal [20,38]. It is the only river that drains from the eastern part of the Kathmandu Valley and confluences with the Manahara River (a major tributary of the Bagmati River) at Jadibuti (Figure 1). This sixth-order river extends up to 18.29 km, covering nearly 97 km$^2$ of watershed areas [39]. The Godawari Khola, Tabyakhusi Khola, and Chakkhu Khola are major tributaries of this river.

The quality of the Hanumante River water is deteriorating as urbanization increases downstream, having lower dissolved oxygen (0-7 mg/L) and higher biological oxygen demand (3.5–79.9 mg/L), chemical oxygen demand (128 mg/L), ammonia (0.4-25 mg/L), and phosphorous (0.09-1.71 mg/L), such that the river water is harmful for domestic purposes [38]. Direct disposal of sewage and solid waste effluent from industries converts the Hanumante River into an open sewer during the dry season [40]. The sole municipal drinking water supply organization, namely, Kathmandu Upatyaka Khanepani Limited (KUKL), cannot fulfill the total water demands of the Kathmandu Valley [41], compelling it to fulfill the water deficit by extracting groundwater from shallow as well as deep aquifers. Meanwhile, Gautam et al. [17] discussed the high possibility of shallow aquifer contamination by polluted river water in the peripheral part of the rivers in the Kathmandu Valley.

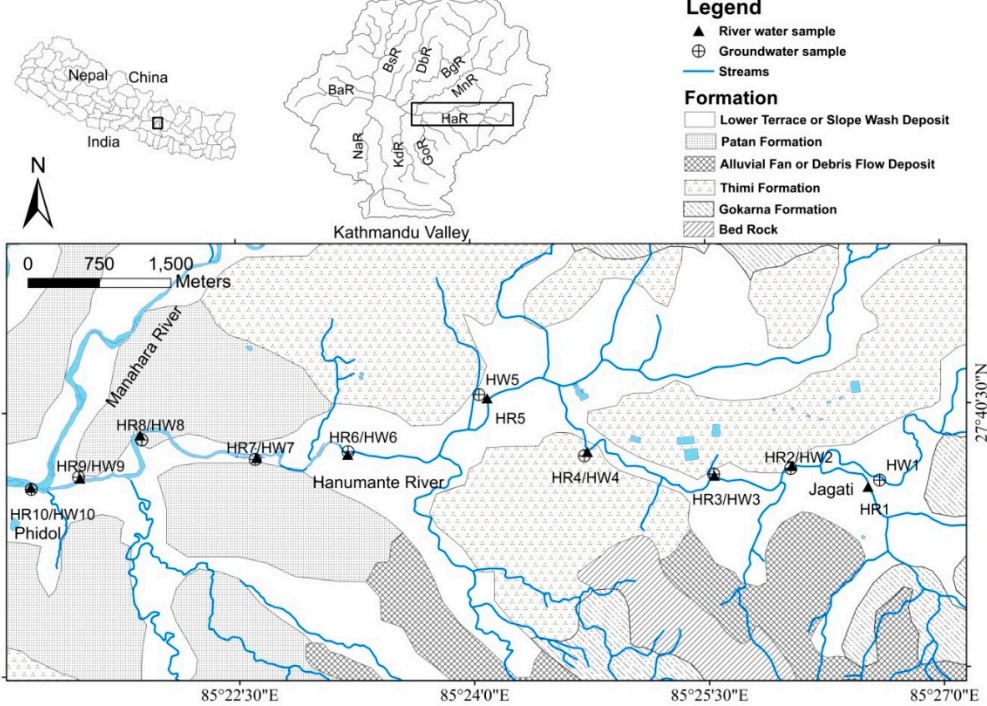

**Figure 1.** Sedimentological map showing study area and sampling locations along the Hanumante River corridor (modified from Yoshida and Igarashi [42], Sakai et al. [43], and Dhital [44]); BsR = Bishnumati, DbR = Dhobi, BgR = Bagmati, MnR = Manahara, HaR = Hanumante, GoR = Godawari, KdR = Kodku, NaR = Nakhhu, and BaR = Balkhu river.

Geologically, Plio-Pleistocene fluvial, fluvio-lacustrine, and fluvio-deltaic sediments comprise the Kathmandu Valley [42,43]. The study area presented in Figure 1 is characterized by four formations, namely, the Gokarna Formation, Patan Formation, Thimi Formation, and Lower Terrace Deposit [44]. The upstream section of the study area is composed of the Gokarna Formation, containing dark brown colored, laminated arkosic sand, silty clay, and peat. The middle section of the study area is covered by the Thimi Formation, which consists of sand, silt, clay, peat, and gravel composed of granite and gneiss derived from the Shivapuri Range. Similarly, the lower section of the study area is dominated by the Patan Formation, which contains deposits of fluvial-lacustrine composed of sand, silt, clay, and peat. The Lower Terrace Deposit along the river corridor consists of micaceous sand, pebbles, and granules [44].

To use chemical and isotopic analysis to investigate any interconnectivity between river and groundwater, samples were collected from 20 locations—10 from rivers and 10 from dug wells (Figure 1) during the wet (August 2017) and dry seasons (February 2018).

### 2.2. FieldMeasurement and Water Sampling

Groundwater was collected from dug wells which are located within 10 to 100 m from the river channel, with the depth ranging between 1.5 and 15.7 m. Sampling was carried out in two seasons—August 2017 (wet season) and February 2018 (dry season)—where the samples were collected from 20 locations, 10 from rivers and 10 from dug wells (Figure 1) in the consecutive seasons, respectively. Water samples were collected in 100 mL polyethylene bottles. Each bottle was rinsed three times with the same water before sample collection. Groundwater samples were collected after removing a quantity of water using an installed hand pump or with the help of rope and a plastic bucket. The collected water samples were stored at −4°C at a laboratory until the chemical and isotope analyses were performed

Additionally, during the sample collection, well depth, water level depth, electrical conductivity (EC), dissolved oxygen(DO), pH, and water temperature were measured at each sampling location.

Well depth was measured using a measuring tape and was verified with the dug well owner. A water depth logger was deployed for the water table measurement. In situ parameters were measured by using portable devices, namely, a DO meter (Mettler Toledo SG3-ELK, Greifensee, Zurich, Switzerland) and a pH/EC meter (Mettler Toledo Duo, Greifensee, Zurich, Switzerland). The location of the water samples is shown in Figure 1, and Table 1 presents the data measured during the field survey.

### 2.3. Chemical and Isotopic Analysis

The Interdisciplinary Center for River Basin Environment, University of Yamanashi (ICRE-UY), Japan provided laboratory facilities to carry out chemical and isotopic analyses. As per the laboratory procedure, collected water samples were first filtered through 0.2 μm filter paper to prepare final samples for further laboratory analyses. The dominant chemical ions, including cations ($Na^+$, $K^+$, $NH_4$-$N^+$, $Ca^{2+}$, and $Mg^{2+}$) and anions ($Cl^-$, $NO_3^-$-N, $PO_4^-$-P, and $SO_4^{2-}$), were determined by using ion chromatography (ICS-1100, Dionex, Waltham, MA, USA with an analytical error of 5%. The bicarbonate ion ($HCO_3^-$) concentrations were measured by using a titration method with 0.01N sulfuric acid.

The stable isotopes of hydrogen (δD) and oxygen ($δ^{18}O$) were analyzed using cavity ring-down spectroscopy (L1102-i, Picarro, Santa Clara, CA, USA). VSMOW (Vienna Standard Mean Ocean Water) is the standard water used to calculate isotopic ratios (δ) of D and $^{18}O$ of water samples. The results were reported in parts per thousand (per mill deviation) with respect to these standards with precision 0.5‰ for δD and 0.1‰ for $δ^{18}O$. The isotopic ratios of hydrogen and oxygen were calculated by using the formula given by Craig [45]:

$$δ = [(R_{sample} − R_{standard})/R_{standard}] × 1000 \text{ (‰)}$$

R is defined as D/H or $^{18}O/^{16}O$ in sampled water ($R_{sample}$) and standard mean ocean water ($R_{standard}$).

### 2.4. Statistical Analysis

Temporal variations of chemical variables were evaluated using a paired t-test for significant difference in parameters [46,47] within a 95% confidence level. Spearman's rho correlation analysis [48] was adopted to establish any relationship among different variables. Hierarchical cluster analysis (HCA) was used to examine any similarity in chemical as well as isotopic composition between river water and groundwater. Cluster analysis is useful in distinguishing water showing similar chemical or isotopic composition from dissimilar ones [37,49,50]. HCA was performed based on Ward's linkage method [51] with squared Euclidean distances as a measure of similarity between samples [11,40]. Statistical Package for Social studies version 25 (SPSS Inc., Chicago, IL, USA) was used for statistical analyses.

## 3. Results

### 3.1. InSitu Parameters

Table 1 presents locations of sample points, the depth of water level at different wells, and the measured in situ parameters in wet and dry seasons. All sample wells had shallow water depth in the wet season, which may imply high recharge and lower extraction rates during wet seasons, with a maximum fluctuation of 3.60 m and a minimum fluctuation of 0.65 m at HW9 and HW6, respectively (Table 1).

In situ parameters, namely, temperature, pH, EC, and DO, in river water and groundwater were measured in wet and dry seasons. The temperature of river water ranged from 22.3 °C to 24.3 °C in the wet season, while in the dry season the range was 14.0–17.9 °C. The temperature range of groundwater was 20.7–24.1°C in the wet season and 13.4–20.5 °C in the dry season (Table 1). The pH value slightly decreased during the dry season in both river water and groundwater (Table 1). EC measured in groundwater ranged from 290 to 934 μS/cm in the wet season, and from 576 to 1323μS/cm in the dry season. Groundwater exhibited higher EC in the dry season relative to the wet season, except at HW1, HW2, and HW4 (Figure 1, Table 1). However, in the case of river water, the value of EC was low (164.2 to 247 μS/cm) in the wet season and abruptly increased by up to eight times (604 to 2060 μS/cm) in the dry season. DO was high in river water during the wet season and abruptly decreased below the value measured in groundwater during the dry season (Table 1).

### 3.2. Hydro-Chemical Parameters

Figure 2 presents spatial and temporal variations of the chemical parameters analyzed from river water and groundwater. $Na^+$, $K^+$, $NH_4^+$-N, $Ca^{2+}$, $Mg^{2+}$, $Cl^-$, $HCO_3^-$, $NO_3^-$-N, $PO_4^-$-P, and $SO_4^{2-}$ are considered as chemical parameters in this study.

$Ca^{2+}$ and $HCO_3^-$ were the most dominant ions of river water in the wet season, with values ranging from 7.3 to 17.0 mg/L and 24.4 to 73.2 mg/L, respectively (Figure 2). The major cations had an order of $Ca^{2+} > Na^+ > K^+ > Mg^{2+}$ in the wet season, which changed to the order $Na^+ > NH_4^+ > Ca^{2+} > K^+ > Mg^{2+}$ in the dry season. Similarly, for anions, $HCO_3^-$ was dominant, followed by $SO_4^{2-}$ and $Cl^-$ in the wet season, and $Cl^-$ and $SO_4^{2-}$ in the dry season. Except for $NO_3^-$-N, all other parameters showed strong significant seasonal variation ($p < 0.01$). Concentrations of all these parameters increased in the dry season, but the rate of increment varied for different parameters. $NH_4^+$-N and $PO_4^-$-P concentration was insignificant (<1 mg/L) in the wet season and significantly increased in the dry season, ranging from 10.3 to 102.9 mg/L (for $NH_4^+$-N) and from 2.4 to 31.7 mg/L (for $PO_4^-$-P). Similarly, concentrations of $Na^+$, $K^+$, $Cl^-$, and $HCO_3^-$ increased by more than ten times than in the wet season (Figure 2). Based on piper plot from the chemical analyses, the Hanumante River can be categorized as Ca-HCO$_3$ type in the wet season and Na-K-HCO$_3$ type in the dry season (Figure 3).

**Table 1.** In situ measured parameters of groundwater and river water in the wet (7 August 2017) and dry (18 February 2018) seasons.

| Sampling ID | N | E | River Bank | Distance from River (m) | Well Depth (m) | Water Level Depth (m) | | | EC (µs/cm) | | pH | | DO (mg/L) | | Water Temp (°C) | |
|---|---|---|---|---|---|---|---|---|---|---|---|---|---|---|---|---|
| | | | | | | Wet | Dry | Difference WLD | Wet | Dry | Wet | Dry | Wet | Dry | Wet | Dry |
| HW1 | 27.66936 | 85.44028 | Right | 100 | 3.3 | 1 | 2.82 | 1.82 | 839 | 797 | 7.03 | 6.40 | 2.05 | 0.44 | 22.3 | 20.5 |
| HW2 | 27.6696 | 85.43617 | Left | 10 | 15.7 | 1 | 1.97 | 0.97 | 934 | 820 | 8.05 | 7.35 | 1.54 | 0.51 | 20.7 | 18.9 |
| HW3 | 27.66915 | 85.42796 | Right | 15 | 8.2 | 0.9 | 4.1 | 3.2 | 784 | 1323 | 7.22 | 6.79 | 1.14 | 0.52 | 23.6 | 19.0 |
| HW4 | 27.67104 | 85.41425 | Left | 60 | 1.5 | 0.6 | 1.74 | 1.14 | 681 | 576 | 7.16 | 6.40 | 2.38 | 2.27 | 22.0 | 13.4 |
| HW5 | 27.67638 | 85.40083 | Right | 80 | 5 | 1 | 3.38 | 2.38 | 290 | 637 | 7.75 | 6.09 | 3.16 | 1.27 | 24.1 | 17.5 |
| HW6 | 27.67166 | 85.38907 | Right | 20 | 2.8 | 1.7 | 2.35 | 0.65 | 520 | 611 | 7.06 | 6.13 | 2.56 | 1.25 | 22.8 | 17.5 |
| HW7 | 27.67103 | 85.37917 | Left | 30 | 7 | 4.3 | 5.8 | 1.5 | 591 | 856 | 7.52 | 6.48 | 2.45 | 1.97 | 22.7 | 17.6 |
| HW8 | 27.673 | 85.36717 | Left | 35 | 5 | 3.6 | 4.5 | 0.9 | 889 | 1192 | 7.25 | 6.39 | 4.10 | 1.82 | 22.6 | 20.4 |
| HW9 | 27.67027 | 85.3619 | Right | 20 | 2.8 | 0.8 | 4.4 | 3.6 | 769 | 964 | 6.73 | 6.28 | 3.00 | 2.08 | 21.0 | 18.3 |
| HW10 | 27.66843 | 85.35531 | Left | 10 | 2.4 | 0.1 | 1.9 | 1.8 | 623 | 1168 | 7.61 | 6.70 | 3.29 | 1.65 | 22.8 | 16.2 |
| HR1 | 27.66722 | 85.44194 | | | | | | | 164.2 | 604 | 8.00 | 7.45 | 6.48 | 5.54 | 22.3 | 14.9 |
| HR2 | 27.66916 | 85.43388 | | | | | | | 171.6 | 1340 | 8.06 | 6.84 | 6.30 | 1.67 | 22.4 | 16.5 |
| HR3 | 27.66833 | 85.42555 | | | | | | | 186.6 | 2010 | 7.22 | 7.02 | 5.50 | 0.51 | 23.0 | 16.6 |
| HR4 | 27.67083 | 85.41222 | | | | | | | 226 | 1780 | 7.78 | 6.88 | 4.43 | 0.67 | 23.0 | 14.0 |
| HR5 | 27.67583 | 85.40166 | | | | | | | 224 | 2060 | 7.85 | 6.75 | 4.59 | 0.55 | 23.9 | 15.6 |
| HR6 | 27.67083 | 85.38666 | | | | | | | 227 | 1924 | 7.81 | 7.00 | 4.26 | 0.24 | 24.3 | 17.2 |
| HR7 | 27.67055 | 85.37694 | | | | | | | 240 | 1998 | 7.96 | 7.00 | 3.99 | 0.32 | 23.5 | 16.7 |
| HR8 | 27.67277 | 85.36472 | | | | | | | 237 | 1955 | 7.80 | 6.99 | 4.03 | 0.15 | 23.5 | 17.1 |
| HR9 | 27.66861 | 85.35805 | | | | | | | 247 | 1960 | 6.70 | 7.00 | 4.05 | 0.07 | 23.0 | 17.9 |
| HR10 | 27.66805 | 85.35305 | | | | | | | 239 | 1985 | 7.96 | 6.98 | 4.25 | 0.44 | 23.4 | 16.4 |

HW = Hanumante well water, HR = Hanumante river water, WLD = water level depth.EC = electrical conductivity, DO = dissolved oxygen.

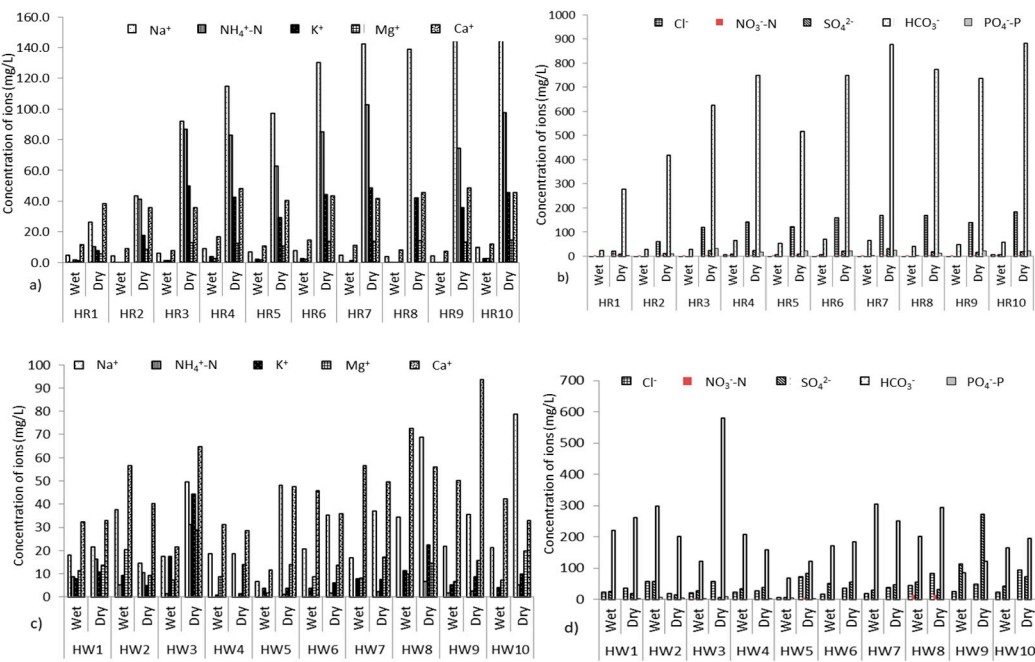

**Figure 2.** Bar diagram showing temporal and spatial variation of chemical parameters in river water (**a**,**b**) and groundwater (**c**,**d**). HR = Hanumante river water and HW = Hanumante groundwater.

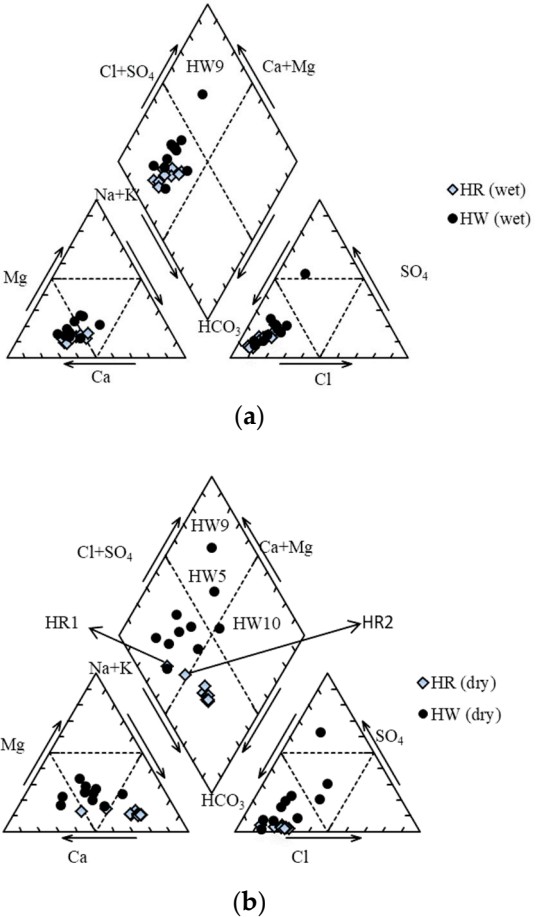

**Figure 3.** Piper diagram of major ions of groundwater and river water in (**a**) wet season (**b**) dry season. HR = Hanumante river water, HW = Hanumante groundwater.

Likewise, $Ca^{2+}$ and $HCO_3^-$ were the most dominant ions of groundwater in both wet and dry seasons. Concentration of $Ca^{2+}$ ranged from 11.7 to 72.5 mg/L in the wet season and from 28.4 to 93.7 mg/L in the dry season. In the case of $HCO_3^-$, 67.1-305.0 mg/L was the range in the wet season, which increased to 122.0–579.5 mg/L in the dry season. Statistical analysis (paired t-test within a 95% confidence level) showed no significant temporal variation in $K^+$, $NH_4^+$-N, $Ca^{2+}$, $HCO_3^-$, $NO_3^-$-N, and $SO_4^{2-}$ in groundwater. However, for $Na^+$, $Mg^{2+}$, and $Cl^-$, temporal variation was significant, with a *p*-value of 0.03. In general, groundwater showed lesser increments in concentration compared with river water during the dry season (Figure 2) and is classified as Ca-HCO$_3$ type in both dry and wet seasons, except at HW5, HW9, and HW10. Water samples collected from HW5 and HW10 changed slightly from Ca-HCO$_3$ (in the wet season) to Ca-SO$_4$, and Na-Cl-SO$_4$, respectively, during the dry season (Figure 3). Groundwater collected from HW9 in both seasons is of Ca-SO$_4$ type.

Determination of water types using a piper diagram suggests the origin of the water [52]. Ca-HCO$_3$ type represents recent infiltration of freshwater, whereas Ca-SO$_4$ and Na-K-HCO$_3$ types indicate water exhibiting simple dissolution or mixing and ion exchange, respectively [53]. Groundwater and river water from the wet season of Ca-HCO$_3$ type thus represents recent rainfall infiltration or runoff as a major contributing source for groundwater recharge and river discharge. Changes in river water type during the dry season reflect the presence of different water sources for river discharge, including direct discharge of untreated sewage from municipal and industrial sources [16].

The Spearman's rho correlation matrix between different chemical parameters in river water and groundwater is presented in Table 2. In river water, DO has a strong negative correlation ($r = -0.60$ to $-0.86$) with all parameters except pH and $NO_3^-$-N. The negative correlation of DO indicates that the presence of a higher concentration of chemical parameters decreases the amount of dissolved oxygen in river water. In contrast, EC has a strong positive correlation with parameters which have a negative correlation to DO, suggesting full dependence of conductivity of river water on dissolved ion concentrations [50]. The higher value of EC and positive correlation with most ions also indicate higher anthropogenic contamination during the dry season [54] (Table 1 and Figure 2). There is also a strong positive correlation of $Na^+$, $K^+$, $NH_4^+$-N, $Ca^{2+}$, $Mg^{2+}$, $Cl^-$, $HCO_3^-$, $PO_4{}^-$P, and $SO_4^{2-}$ between each other ($r = 0.67$ to 0.98), indicating that the river water is highly influenced by anthropogenic pollution [11] such as direct discharge of municipal and industrial sewage and leachate of solid waste disposal near the river channel during the dry season [40]. Strong positive correlation of $PO_4^-$-P with $SO_4^{2-}$, $NH_4^+$-N, and $NO_3^-$-N also suggests the influence of fertilizer and pesticides used in the cultivated land of river peripheral areas [54–56].

Correlations of chemical parameters of groundwater are similar to those of river water. As in the case of EC, it has strong positive correlation with $Na^+$, $K^+$, $NH_4^+$-N, $Ca^{2+}$, $Mg^{2+}$, $Cl^-$, and $PO_4^-$-P. Strong positive correlation between $Na^+$, $Mg^{2+}$ and $Cl^-$ (0.94) in groundwater indicates the influence of anthropogenic activities [57] because there was no evidence of halite deposits in the study area [58]. The negative correlation between $NH_4^+$-N and $NO_3^-$-N represents nitrification of $NH_4^+$-N into $NO_3^-$-N [59]. Positive correlation of $K^+$, $NH_4^+$-N, and $PO_4^-$-P represents agricultural impact from the surrounding cultivated land.

Table 2. Correlation matrix of different chemical parameters.

**(a) River Water**

| Parameters | DO | EC | pH | $Na^+$ | $NH_4^+$-N | $K^+$ | $Mg^{2+}$ | $Ca^{2+}$ | $Cl^-$ | $NO_3$-N | $PO_4$-P | $SO_4^{2-}$ | $HCO_3^-$ |
|---|---|---|---|---|---|---|---|---|---|---|---|---|---|
| DO | 1.00 | | | | | | | | | | | | |
| EC | −0.86 | 1.00 | | | | | | | | | | | |
| pH | 0.64 ** | −0.72 | 1.00 | | | | | | | | | | |
| $Na^+$ | −0.77 | 0.76 ** | −0.54 | 1.00 | | | | | | | | | |
| $NH_4^+$-N | −0.83 | 0.87 ** | −0.61 | 0.87 ** | 1.00 | | | | | | | | |
| $K^+$ | −0.72 | 0.80 ** | −0.49 | 0.92 ** | 0.92 ** | 1.00 | | | | | | | |
| $Mg^{2+}$ | −0.78 | 0.81 ** | −0.54 | 0.98 ** | 0.90 ** | 0.95 ** | 1.00 | | | | | | |
| $Ca^{2+}$ | −0.73 | 0.74 ** | −0.49 | 0.94 ** | 0.82 ** | 0.87 ** | 0.92 ** | 1.00 | | | | | |
| $Cl^-$ | −0.77 | 0.79 ** | −0.57 | 0.98 ** | 0.91 ** | 0.95 ** | 0.98 ** | 0.92 ** | 1.00 | | | | |
| $NO_3$-N | 0.14 | −0.30 | 0.30 | 0.15 | −0.09 | 0.13 | 0.12 | 0.02 | 0.13 | 1.00 | | | |
| $PO_4$-P | −0.61 | 0.93 ** | −0.37 | 0.67 * | 0.89 ** | 0.88 ** | 0.69 * | 0.40 | 0.66 * | 0.70 * | 1.00 | | |
| $SO_4^{2-}$ | −0.70 | 0.77 ** | −0.50 | 0.92 ** | 0.89 ** | 0.98 ** | 0.94 ** | 0.88 ** | 0.93 ** | 0.12 | 0.75 ** | 1.00 | |
| $HCO_3^-$ | −0.85 | 0.86 ** | −0.62 | 0.92 ** | 0.92 ** | 0.90 ** | 0.95 ** | 0.90 ** | 0.94 ** | −0.05 | 0.68 * | 0.89 ** | 1.00 |

**(b) Groundwater**

| Parameters | DO | EC | pH | $Na^+$ | $NH_4^+$-N | $K^+$ | $Mg^{2+}$ | $Ca^{2+}$ | $Cl^-$ | $NO_3$-N | $PO_4$-P | $SO_4^{2-}$ | $HCO_3^-$ |
|---|---|---|---|---|---|---|---|---|---|---|---|---|---|
| DO | 1.00 | | | | | | | | | | | | |
| EC | −0.39 | 1.00 | | | | | | | | | | | |
| pH | 0.36 | −0.18 | 1.00 | | | | | | | | | | |
| $Na^+$ | −0.26 | 0.66 ** | −0.53 | 1.00 | | | | | | | | | |
| $NH_4^+$-N | −0.72 | 0.74 ** | −0.26 | 0.38 | 1.00 | | | | | | | | |
| $K^+$ | −0.38 | 0.75 ** | −0.03 | 0.42 | 0.63 ** | 1.00 | | | | | | | |
| $Mg^{2+}$ | 0.00 | 0.69 ** | −0.46 | 0.78 ** | 0.57 ** | 0.40 | 1.00 | | | | | | |
| $Ca^{2+}$ | 0.10 | 0.51 * | −0.07 | 0.55 * | 0.22 | 0.42 | 0.42 | 1.00 | | | | | |
| $Cl^-$ | −0.32 | 0.71 ** | −0.54 | 0.94 ** | 0.44 | 0.47 * | 0.81 ** | 0.49 * | 1.00 | | | | |
| $NO_3$-N | 0.30 | −0.10 | −0.28 | 0.11 | −0.45 | 0.07 | −0.10 | 0.14 | 0.18 | 1.00 | | | |
| $PO_4$-P | −0.19 | 0.74 * | 0.69 | 0.14 | 0.64 | 0.64 | 0.25 | 0.50 | 0.00 | −0.54 | 1.00 | | |
| $SO_4^{2-}$ | 0.25 | 0.12 | −0.36 | 0.54 * | −0.21 | −0.12 | 0.24 | 0.45 * | 0.47 * | 0.19 | −0.48 | 1.00 | |
| $HCO_3^-$ | −0.33 | 0.48 | 0.12 | 0.26 | 0.45 * | 0.48 * | 0.49 * | 0.36 | 0.27 | −0.21 | 0.55 | −0.33 | 1.00 |

* Correlation is significant at the 0.05 level (two-tailed); ** Correlation is significant at the 0.01 level (two-tailed), DO = dissolved oxygen, EC = Electrical conductivity, $NH_4^+$-N = ammonium nitrogen, $NO_3^-$-N = nitrate nitrogen, $PO_4^-$-P = phosphate phosphorous.

### 3.3. Isotopic Composition

Fewer studies regarding isotopic analysis of meteoric water have been carried out on a local scale [60–62]. The local meteoric water line (LMWL), established by Gajurel et al. [60] and Giri [61], has a similar slope and intercept value as the global meteoric water line (GMWL) reported by Craig [45]. In this study, GMWL is used as a reference of meteoric water. The $\delta^{18}$O verses $\delta$D plot (Figure 4) presents variations in stable isotopic compositions of groundwater and river water during both wet and dry seasons.

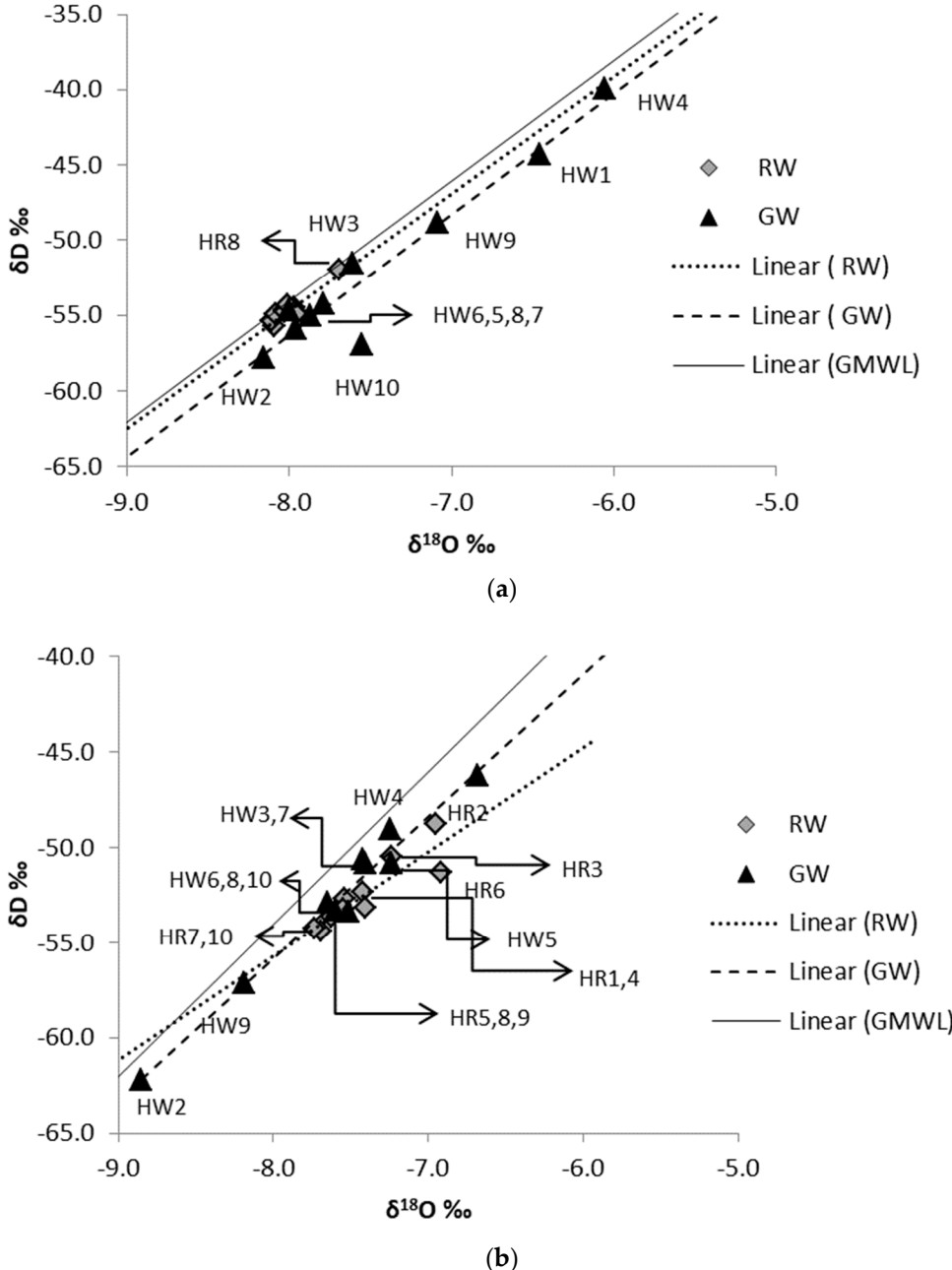

**Figure 4.** $\delta$D verses $\delta^{18}$O plots of groundwater and river water in (**a**) wet season and (**b**) dry season. HW = Hanumante groundwater, HR = Hanumante river water. GMWL represents global meteoric water line [45]; linear (RW) = trend line of river water; linear (GW) = trend line of groundwater.

During the wet season, almost all river samples showed similar isotopic compositions of $\delta^{18}$O (−8.0 to −8.1‰) and $\delta$D (−54.3 to −55.7‰) except HR8, which had a heavier isotopic composition

(Figures 1 and 4). Composition of isotopes slightly changed to heavier with wider value range during the dry season, showing a range from −6.91 to −7.73‰ for $\delta^{18}$O and −48.75 to −54.41‰ for $\delta$D. Research work by Yang et al. [63] in the Jiulong River also presents a narrow and wider value range for the wet and dry season, respectively. Wet season river samples are plotted near the GMWL and have a similar slope and intercept ($R^2$ = 0.86) as the GMWL (Figure 4a), indicating recent meteoric water as a major source for river discharge [4,8,34,58]. Ca-HCO$_3$ water type, defined from piper diagram and lighter isotopic composition of rainfall during wet season [62,64], also suggests a similar water source. However, there was no evidence of evaporation in a previous study in the Kathmandu Valley [58,60]; dry season river samples plotted below the GMWL with lower slope (slope = 5.46 with $R^2$ = 0.79; Figure 4b) as compared with the GMWL may indicate a possibility of evaporation [26,65].

Large spatial variation of $\delta^{18}$O and $\delta$D in water from dug wells was observed in both wet and dry seasons (Figure 4b). The $\delta^{18}$O of groundwater ranged from −6.1‰ to −8.2‰ in the wet season and from−6.68‰ to −8.85‰ in the dry season. However, the overall range of $\delta^{18}$O was similar in both seasons, with samples from HW3, HW5, HW6, HW7, HW8, and HW10 showing lighter composition in the wet season. In both seasons, HW1 had the heaviest isotopic value, whereas HW2 had the lightest isotopic composition. Groundwater samples plot below the GMWL, with a similar slope ($R^2$ > 0.9) in both wet and dry seasons (Figure 4b), indicating recent meteoric water as a major recharge source for these dug wells [12]. Although meteoric water is a major source for dug wells, spatial variation is noticeable in groundwater since isotopic composition of precipitation is dependent on rainfall amount, elevation, and source of water vapor of rainfall [62,64,66].

## 3.4. Clustering of River Water and Groundwater

Isotopic composition is a reliable source to identify recharge sources of groundwater, while the concentration of Na$^+$ and Cl$^-$ can be used as an indicator of the presence of contamination through increased urbanization [36]. Thus, for HCA, $\delta$D, $\delta^{18}$O, Na$^+$, and Cl$^-$ are used as major parameters to identify similarity or interconnection between river water and groundwater [4,30]. All 20 water samples from river water and groundwater of both seasons are managed separately for HCA. In the wet season, three clusters, namely A, B, and C, are observed in a dendrogram (Figure 5a). Cluster A consists of the water samples with lower concentration of Na$^+$, and Cl$^-$. It includes one dug well site (HW5) with all river sites and indicates minor influence of sewage discharge in wet season river water. The clusters B and C consist entirely of groundwater locations (Figure 5a) with higher concentration of Na$^+$ and Cl$^-$ and different isotopic compositions compared with that of river water locations.

Two major clusters, D and E, are categorized from a dendrogram of the dry season (Figure 5b). Cluster D consists only of river sites (eight sites), whereas Cluster E contains a combination of river and groundwater sites, indicating similarity in selected parameters [56,67]. Cluster E has two subgroups, E1 and E2. Six dug well sites and one river site (HR1) are contained in Cluster E1. Similarly, four dug well sites and one river site (HR2) are grouped in Cluster E2. These two river samples, which are clustered with groundwater in Cluster E, are located at the uppermost section of the river (Figure 1), having lower concentrations of Na$^+$ and Cl$^-$ and classified as Ca-HCO$_3$ type. However, in the case of the river sites of Cluster D, they have the highest concentration of Na$^+$ and Cl$^-$, and it is categorized as aNa-K-HCO$_3$ water type, which indicates pronounced effects of sewage discharge. Continued discharge of sewage in river water can develop an organic clogging layer on the riverbed [2], which will drastically reduce water exchange from the river channel to groundwater [3]. The absence of groundwater samples grouping with river samples in Cluster D may indicate the possibility of clogging layer formation within the river bed. However, the clusters combining river water and groundwater, Cluster A from the wet season (Figure 5a) and Cluster E from the dry season (Figure 5b), show similarity in selected parameters and indicate interconnection between river and groundwater [9,11,37].

However, the current research has a limitation in the methodological approach where the river channel stage could not be developed comparing relative river water and groundwater level.

The HCA along with hydraulic head between river water and groundwater can give a clearer view of interconnection between river and groundwater.

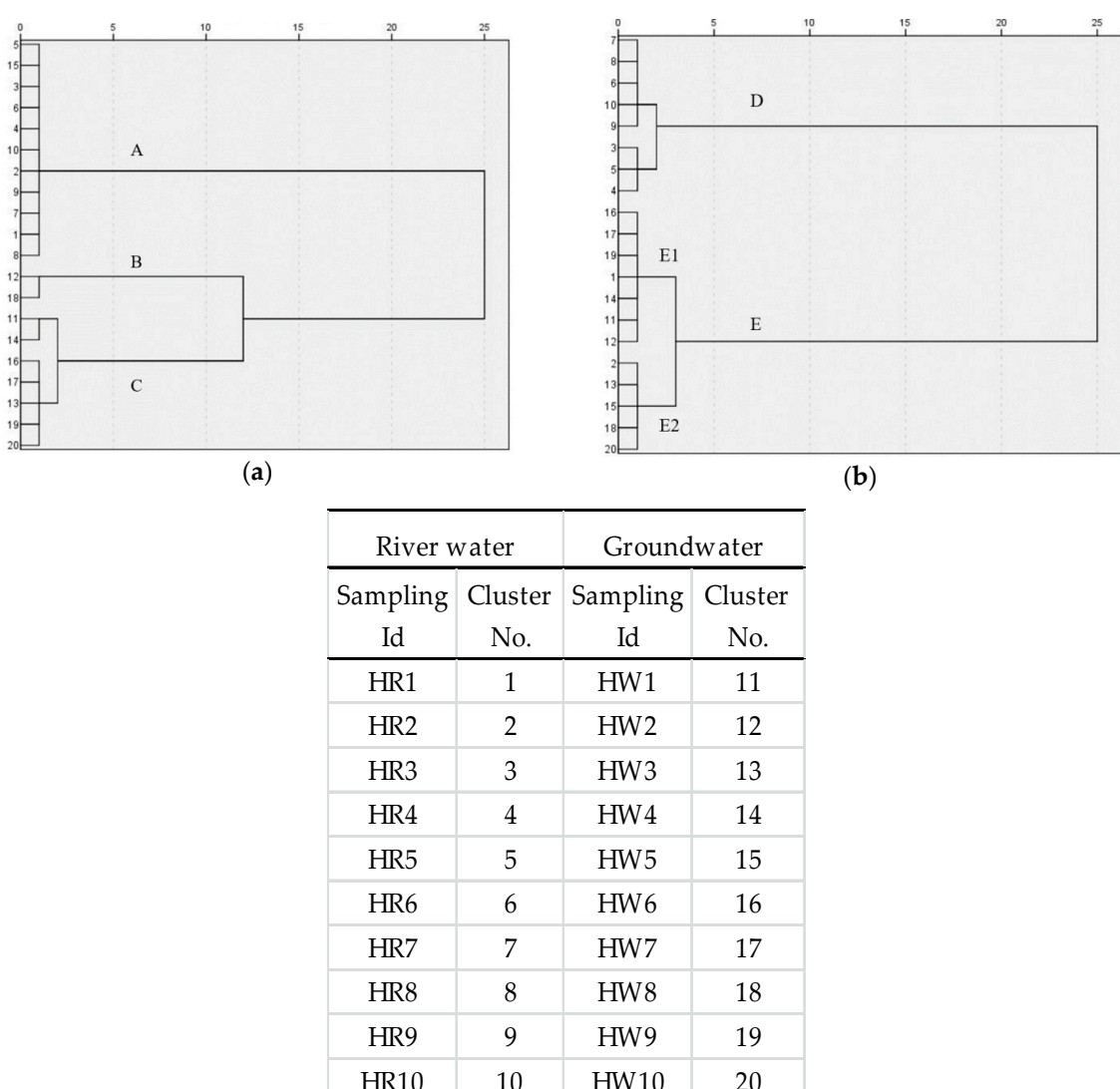

|  | River water |  | Groundwater |  |
|---|---|---|---|---|
| Sampling Id | Cluster No. | Sampling Id | Cluster No. |
| HR1 | 1 | HW1 | 11 |
| HR2 | 2 | HW2 | 12 |
| HR3 | 3 | HW3 | 13 |
| HR4 | 4 | HW4 | 14 |
| HR5 | 5 | HW5 | 15 |
| HR6 | 6 | HW6 | 16 |
| HR7 | 7 | HW7 | 17 |
| HR8 | 8 | HW8 | 18 |
| HR9 | 9 | HW9 | 19 |
| HR10 | 10 | HW10 | 20 |

(c)

**Figure 5.** Dendrogram based on hierarchical clustering (Ward's method) in (**a**) wet season and (**b**) dry season. (**c**) Relation of Hanumante river water (HR) and Hanumante groundwater (HW) with cluster number in dendrogram.

## 3.5. Identifying Areas of River Water and Groundwater Interconnection

Results from HCA imply interconnection between river water and groundwater in both wet and dry seasons. Cluster A from the wet season combines all river water with one groundwater (HW5), showing the similar water type Ca-HCO$_3$. HW5 has shallow well depth (5 m) with water level depth at 1 m. It is located in permeable lower terrace deposits (Figure 1), generally used for cultivation. Many flooding events were recorded for the Hanumante River [68], with a major one recently in July 2018 [69] which inundated the entire area from Jagati to Madhyapur Thimi, along with HW2, HW5, and HW6 (Figure 1). However, HW5 is the only groundwater sample that was grouped with all river sites in Cluster A (Figure 5); HR5 is closest site to HW5 (80 m away; Figure 1). Identical isotopic compositions of δ$^{18}$O (−8.0‰) and δD (−54.7‰) observed in HW5 and HR5 suggest possible recharge of groundwater in HW5. Similar concentration of Na$^+$ and Cl$^−$ with HR5, along with drastic dilution

of these concentrations (Figure 2) at HW5, supports bank infiltration [4,18,19,70] as the mechanism recharging HW5 during the wet season (August of 2017). Similar types of results are presented by previous research conducted in developing countries for local scale [36] as well as for regional scale [37], indicating river water can recharge nearby groundwater.

In the dry season, dug wells HW1, HW2, HW4, HW6, HW7, and HW9 are grouped with the river site HR1 in Cluster E1, whereas dug wells HW3, HW5, HW8, and HW10 grouped with HR2 are clustered in E2. However, Cluster E from the dry season (Figure 5b) is suggestive of possible interaction between river water (HR1 and HR2) and groundwater. As HR1 and HR2 are located at the uppermost section of the river channel compared with grouped groundwater sites (Figure 1), there is a higher possibility of downstream groundwater recharge by HR1 and HR2 in the dry season. But differences in isotopic compositions of river water and nearby groundwater samples (Figure 4b) imply the possibility of other recharge sources besides river water. Further, groundwater samples exhibited similar trends of isotopic composition as that of the GMWL, suggesting dry season rainfall as one of the alternative sources of recharge for these dug wells.

Dug wells HW2, HW3, HW9, and HW10 are located within 20 m of HR2, HR3, HR9, and HR10, respectively (Figure 1). However, no single cluster formed includes this entire sampling site. These results indicate that interaction between river water and groundwater not only depends on distance from the river channel to groundwater sites, it also depends on well depth, water level depth, topography of well location, water level in river channel, and sedimentological formation of the areas.

*3.6. River Water Contribution to Groundwater*

Dug wells recharged by river water have a mixture of water, including river water (RW) and original groundwater (GW). During the study period, only HW5 showed interconnection with river water during the wet season. The proportion of river water to groundwater can be estimated using a mass balance approach equation [18,36]:

$$f = [(CS − CGW)/(CRW − CGW)] \times 100\%$$

where CS is the $Cl^-$, $\delta D$, or $\delta^{18}O$ (mg/L or ‰) of mixed water (HW5 in the wet season); CRW is the $Cl^-$, $\delta D$, or $\delta^{18}O$ of river water (HR5 in the wet season), and CGW is the $Cl^-$, $\delta D$, or $\delta^{18}O$ of original groundwater (HW5 in the dry season) which has not been influenced by river water recharge.

The calculated fractional contribution is nearly 100% for $Cl^-$, $\delta D$, or $\delta^{18}O$, indicating the contribution of river water (HR5) to recharge HW5 during the wet season (August 2017). This is the only well which had an identical isotopic value as in HR5 with much diluted water during the wet season.

The existence of interconnection between the Hanumante River and shallow groundwater shows that the surrounding shallow groundwater is contaminated by polluted river water. As river water deterioration continues with increasing urbanization, groundwater contamination is expected. It is very important to maintain river water quality for the improvement of peripheral shallow groundwater quality.

## 4. Conclusions

This study analyzes chemical and isotopic compositions of river water and groundwater to investigate interconnectivity between river water and shallow aquifers. Hydro-chemical parameters of river water exhibit significant temporal variations compared with groundwater. Groundwater and river water reveal a Ca-$HCO_3$ type with similar trends of isotopic composition as that of the GMWL, confirming freshwater as a major source for groundwater recharge and river discharge during the wet season. Water types shifted from Ca-$HCO_3$ to Na-K-$HCO_3$, and strong positive correlations between $Na^+$, $K^+$, $NH_4^+$-N, $Ca^{2+}$, $Mg^{2+}$, $Cl^-$, $HCO_3^-$, $PO_4$-P, and $SO_4^{2-}$ indicate anthropogenic activities as a major source of contamination in dry season river water. Slight deviations of isotopic compositions

from GMWL with a lower slope may suggest the possibility of evaporation in river water during the dry season.

Clusters formed by combinations of river water and groundwater, both in wet (Cluster A) and dry (Cluster E) seasons, indicate the presence of interconnectivity between river water and shallow groundwater. The wet season included one groundwater sample with all the river samples, including one well which was near the riverside (within 80 m). Identical isotopic composition and similar concentrations of $Na^+$ and $Cl^-$ in groundwater and river water suggest almost 100% of water recharge through bank infiltration comes from river to shallow groundwater during the wet season. Grouping of two upstream river samples with all downstream shallow groundwater implies that upstream river water is one of the sources of recharge for downstream shallow groundwater during the dry season.

This research concluded that polluted river water is one of the contamination sources for shallow groundwater around riverside areas. Such types of studies can be applied in other highly contaminated rivers to give a clear view about the contamination source for river peripheral groundwater. Higher extents of similar studies in rivers of the most urbanized areas are very important for groundwater management of the river peripheral area.

**Author Contributions:** T.N., N.K.T. and S.G. supervised this work. R.B. designed the field work for sample collection; R.B., T.N., and B.M.S. were involved in lab analyses; R.B. performed statistical analysis and prepared the manuscript; S.G., T.N., and B.M.S. revised the manuscript. All authors have read and agreed to the published version of the manuscript.

**Funding:** This research was funded by University Grant Commission (UGC), Nepal (Award No. 73/74/S & T-06).

**Acknowledgments:** The authors are grateful to ICRE-UY, Japan for providing lab facilities and Ramesh Raj Pant for his support in statistical discussions. The authors would like to acknowledge the support of Science and Technology Research Partnership for Sustainable Development (SATREPS), Japan International Co-operation Agency (JICA), Japan Science and Technology Agency (JST), and Grants-in-Aid for Scientific Research (KAKENHI No. 18K11617).

**Conflicts of Interest:** The authors declare no conflict of interest.

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
