# Peer review of "Identifying Groundwater and River Water Interconnections Using Hydrochemistry, Stable Isotopes, and Statistical Methods in Hanumante River, Kathmandu Valley, Central Nepal"

_water, doi:10.3390/w12061524_

Round 1

Reviewer 1 Report

The submitted paper investigates possible interconnectivity between the highly polluted water of the Hanumante River and the shallow aquifers in the peripheral part of the river. In this process well known methodology is used including field measurement and water sampling, chemical and isotopic analysis of collected samples, and classical statistical analysis of the laboratory results. This paper presents a case study and it does not contain any methodological novelty. It is too much concentrated to a contribution to local water management, so it is more appropriate for a local journal in Nepal. In my opinion, it should be reformulated in order to obtain an international interest. It implies (1) international context, examples and references in introduction, (2) an appropriate discussion, and (3) conclusions that can be applied to other similar study areas.

Author Response

Response to Reviewer 1:

The submitted paper investigates possible interconnectivity between the highly polluted water of the Hanumante River and the shallow aquifers in the peripheral part of the river. In this process well known methodology is used including field measurement and water sampling, chemical and isotopic analysis of collected samples, and classical statistical analysis of the laboratory results. This paper presents a case study and it does not contain any methodological novelty. It is too much concentrated to a contribution to local water management, so it is more appropriate for a local journal in Nepal. In my opinion, it should be reformulated in order to obtain an international interest. It implies (1) international context, examples and references in introduction, (2) an appropriate discussion, and (3) conclusions that can be applied to other similar study areas.

Response:

Thank you very much for reviewing our manuscript and offering valuable comments on our manuscript. We have revised our manuscript following your comments and suggestions. The modified parts are highlighted by yellow color in new manuscript.

(1) International context, examples and references in introduction,

Response: Thank you very much for your suggestions. We added more examples of international context in introduction part (Line 34-37, 43-56,67-78) with scientific research paper as a reference added in Line No. 423-424, 425-426,435-437, 443-445, 454-455, 456-459, 510-512

(2) Appropriate discussion

Response: Thank you very much for your suggestions. We added more discussion on section of isotopic composition, clustering of river water and groundwater and identifying areas of interconnection within lines 271-274, 299-301, 305-307, 318-322, 326-329, 350-351, 355-358.

(3) Conclusions that can be applied to other similar study areas.

Response: Thank you very much for your suggestions. We added application of similar studies in lines 407-409.

Reviewer 2 Report

Dear Authors,

in the following you will find several minor revisions, which may contribute to improve your paper.

Overall comments

The manuscript is well written and well stuctured. Please check once again for spelling. There are some small spelling Errors.

Abstract

Line 17: "… one of the most polluted rivers"--> Please add where (in the world, on Asia, in the Himalaya Region, etc)

Line 27: "...that shallow groundwater found near rivers, is contaminated by polluted river water" --> What Kind of contamination do you mean here (chemical, microbiological, etc)?

Introduction

Line 62: "... such that the river water is harmful for domestic purposes". --> This is a fundamental Statement. I suggest to give here the concentration ranges.

Line 72: "... where many temples of archeological importance are located" 

--> Is this Information of importance for the presented work? Are there a lot of tourists and the Population increases during certain seasons?

Line 78 to 84: Description of the stratigraphies --> Can you add a schematic geological profile of the described area?

Line 92: "... during the wet and dry Season". --> I suggest to add already the year of sampling here. I suggest to add some meteorological data here, i.e. Volume of precipitation in the wet and dry Season, comparison to Long time average values to show the representativity of the sampled Seasons.

Materials and Methods

How did you pump the water from the wells? Were there pumps installed or did you use an own pump or did you use a bailer?

How did you take the river water samples?

How did you measure the on-site parameters in the river and groundwater?

Line 104: "... at -4°C" --> Did you really freeze the samples? If so, why? or did you store them at +4°C?

Line 107: "...DO, pH and water temperature were measured at each sampling Location". --> At which pumping/flow rate, did you monitor time series, etc. Please explain a bit more in Detail.

End: Did you also measure DOC/TOC and/or BOD and COD? If not, why not? These would be essential Parameters for anthropogenic contamination and a good possibility for a literature comparison (cf. reference 28).

Results

Are the sampled wells only Observation wells or used groundwater production wells.

Line 148: …" and 13.4 - 20.5°C in the dry Season" --> How can you explain the warm temperature? Is there a heat source (Industry, heat pump, etc.) close by?

Line 154: "DO is relatively high" --> What does relatively mean? Please avoid such qualitative statements

Line 157, Figure 2: I suggest to add the concentrations of the chemical Parameters also in a table, comparable with Table 1. 

Line 171: "... can be categorized as …" --> Which classification did you use here?

Table 1: I suggest to add a column with the sampling date.

Please make sure that you Always use the same number of decimal places (see e.g. EC HR1-HR3 or the pH-values).

Line 229: "The negative correlation of DO indicates that the presence of a higher concentration of chemical paramters decreases the amount of DO in river water."--> Are These really the parameters and processes, which consume the DO? Or is it more the corresponding organics? 

Line 281: "indicate meteoric water"--> What else shell it be? I suggest to write recent meteoric water

Conclusion

Line 368:"... evaporation in dry river water"--> What does this mean?

Author Response

Response to Reviewer 2:

Overall comments

The manuscript is well written and well structured. Please check once again for spelling. There are some small spelling errors.

Response:

Thank you very much for reviewing our manuscript, offering valuable comments and for your judgment on our manuscript. We have revised our manuscript following your comments and suggestions.

Abstract

Line 17: "… one of the most polluted rivers"--> Please add where (in the world, on Asia, in the Himalaya Region, etc)

Response: Thank you very much for your suggestion. We removed this sentence from Line 17, as this river is of local scale and cannot be consider regarding Asia or Himalaya Region. Instead of this, we used “river and shallow groundwater interconnection in urbanized areas of the Kathmandu Valley, Nepal”

Line 27: "...that shallow groundwater found near rivers, is contaminated by polluted river water" --> What Kind of contamination do you mean here (chemical, microbiological, etc)?

Response: Thank you very much for your question. “Chemically contaminated” is added in Line 27

Introduction

Line 62: "... such that the river water is harmful for domestic purposes". --> This is a fundamental Statement. I suggest to give here the concentration ranges.

Response: Thank you very much for your suggestions. The value ranges of each parameter are added in Line 86-88.

Line 72: "... where many temples of archeological importance are located" 

--> Is this Information of importance for the presented work? Are there a lot of tourists and the Population increases during certain seasons?

Response: Thank you very much for your question. The Hanumante River is basically flowing through the Bhaktapur City, which is one of tourist’s attractions in Nepal, where most of the world heritages in Nepal are located.

Line 78 to 84: Description of the stratigraphies --> Can you add a schematic geological profile of the described area?

Response: Thank you very much for your suggestion. We want to apologize for being unable to prepare geological profile of this area. But for the future studies, we will be preparing such profiles in the study area.

Line 92: "... during the wet and dry Season". --> I suggest to add already the year of sampling here.

Response: Thank you very much for your suggestion. We added year of sampling in Line no. 122.

Materials and Methods

How did you pump the water from the wells? Were there pumps installed or did you use an own pump or did you use a bailer?

Response: Thank you very much for your question. Hand pumps are installed at some site but we mostly use hand held method i.e. rope and plastic bucket to collect water samples from dug well.

How did you take the river water samples?

Response: Thank you very much for your question. Actually we collect water sample using rope and plastic bucket from the central run under the bridge. But at some locations where bridges are not available, we collect directly from river channel.

How did you measure the on-site parameters in the river and groundwater?

Response: Thank you very much for your question. We first collected water sample in a big plastic bucket form river and groundwater and then we dip portable DO meter and EC meter in the water contain bucket.

Line 104: "... at -4°C" --> Did you really freeze the samples? If so, why? or did you store them at +4°C?

Response: Thank you very much for pointing out our mistakes. Actually we store water samples in -4°C because higher temperature might effects on stable isotope and chemicals.

Line 107: "...DO, pH and water temperature were measured at each sampling Location". --> At which pumping/flow rate, did you monitor time series, etc. Please explain a bit more in Detail.

Response: Thank you very much for your question. The selected all wells for this research are of dug well type. So the water samples are mostly collected by using rope and plastic bucket. But before sample collection, we remove some quantity of water sample such that contamination on upper surface of water can remove.

End: Did you also measure DOC/TOC and/or BOD and COD? If not, why not? These would be essential Parameters for anthropogenic contamination and a good possibility for a literature comparison (cf. reference 28).

Response: Thank you very much for your question. But we do not measure DOC/TOC and/or BOD and COD. Actually this research is designed for interconnectivity and we only focuses on isotope and other chemical ions.

Results

Are the sampled wells only Observation wells or used groundwater production wells.

Response: Sampled wells are groundwater production wells.

Line 148: …" and 13.4 - 20.5°C in the dry Season" --> How can you explain the warm temperature? Is there a heat source (Industry, heat pump, etc.) close by?

Response: Thank you very much for your question. Except HW4, Temperature of groundwater range from 16 to 20°C (Table 1). The HW4 lies in sheltered area at where direct sunlight cannot be reached at this site. May be this will be possible reason for lower temperature as compared to other sites.

Line 154: "DO is relatively high" --> What does relatively mean? Please avoid such qualitative statements

Response: Thank you very much for your suggestion. We removed “relatively” word.

Line 157, Figure 2: I suggest to add the concentrations of the chemical Parameters also in a table, comparable with Table 1. 

Response: We are confused little bit. Do you mean to substitute Figure 2 by Table or we need to keep both table and Figure 2? So in this new manuscript, only Figure 2 is includes and Table is provided in supplementary data.

Line 171: "... can be categorized as …" --> Which classification did you use here?

Response: Thank you very much for your question. We used piper diagram for water classification and add “Piper diagram” in Line no.197.

Table 1: I suggest to add a column with the sampling date.

Response: Thank you very much for your suggestion. All samples were collected at same day for each season. So sampling date is added in the title of the Table 1.

Please make sure that you Always use the same number of decimal places (see e.g. EC HR1-HR3 or the pH-values).

Response: Thank you very much for your suggestion. We made correction in Table 1.

Line 229: "The negative correlation of DO indicates that the presence of a higher concentration of chemical paramters decreases the amount of DO in river water."--> Are These really the parameters and processes, which consume the DO? Or is it more the corresponding organics? 

Response: Thank you very much for your question. These are corresponding organics which can decrease DO.

Line 281: "indicate meteoric water"--> What else shell it be? I suggest to write recent meteoric water

Response: Thank you very much for your suggestion. We added “recent” in Line no 309.

Conclusion

Line 368:"... evaporation in dry river water"--> What does this mean?

Response: Thank you very much for your question. We rewrite “evaporation in river water during dry season”.

Reviewer 3 Report

The paper is robust and could be published after revision based on the following comments

  1. Improve introduction with studies relevant to the subject of the paper with discussion on the various methods (differences and similarities with the methods used in this study)
  2. Part of the introduction contains description of the study area. Remove this and make a new section "Study area" in the Methods section.
  3. Enlarge Figs5A,B and substitute numbers with the codes in the table.
  4. Remove discussion from the results and make a new discussion section. In the discussion section add comments related to limitations in the methodological approach.

Author Response

Response to Reviewer 3:

The paper is robust and could be published after revision based on the following comments

  1. Improve introduction with studies relevant to the subject of the paper with discussion on the various methods (differences and similarities with the methods used in this study)
  2. Part of the introduction contains description of the study area. Remove this and make a new section "Study area" in the Methods section.
  3. Enlarge Figs5A,B and substitute numbers with the codes in the table.
  4. Remove discussion from the results and make a new discussion section. In the discussion section add comments related to limitations in the methodological approach.

Response:

Thank you very much for reviewing our manuscript, offering valuable comments and for your judgment on our manuscript. We have revised our manuscript following your comments and suggestions. The modified parts are highlighted by yellow color in new manuscript.

  1. Improve introduction with studies relevant to the subject of the paper with discussion on the various methods (differences and similarities with the methods used in this study)

Response: Thank you very much for your suggestion. We try to improve the introduction part including research work related to present study. The added parts are highlighted by yellow color in the introduction part.

  1. Part of the introduction contains description of the study area. Remove this and make a new section "Study area" in the Methods section.

Response: Thank you very much for your suggestion. We removed such part from introduction and added new section "Study area" in the Methods section form Line no. 109-122

  1. Enlarge Figs5A,B and substitute numbers with the codes in the table.

Response: We added code in Figure 5a and 5b.

  1. Remove discussion from the results and make a new discussion section. In the discussion section add comments related to limitations in the methodological approach.

Response: Thank you very much for your suggestion. We want to apologize that we did not separate discussion from result in this paper but we added some discussion including limitation of methodological approach in Line no. 341-344.

Round 2

Reviewer 1 Report

No additional comments.

Author Response

Thank you for your good comments and suggestions to improve our manuscript.

Takashi NAKAMURA.

Reviewer 3 Report

My only final comment before accepting the manuscript is the following:

Remove the last two paragraphs (except lines 95-100) of the introduction + Figure 1 and move it to the section of study area. This text has no place in the introduction.

Create a new last paragraph of the introduction (using the previous lines 95-100) with the aim of your study. The last paragraph of introduction should give a brief description about what you are going to do in this study based on the general topics that you previously described. If it is not possible to connect this paragraph with the previous text, means that the previous text of your introduction is not properly formed to introduce the aim of your study in a single paragraph and needs revisions.

Author Response

Response to Reviewer 3:

My only final comment before accepting the manuscript is the following:

Remove the last two paragraphs (except lines 95-100) of the introduction + Figure 1 and move it to the section of study area. This text has no place in the introduction.

Create a new last paragraph of the introduction (using the previous lines 95-100) with the aim of your study. The last paragraph of introduction should give a brief description about what you are going to do in this study based on the general topics that you previously described. If it is not possible to connect this paragraph with the previous text, means that the previous text of your introduction is not properly formed to introduce the aim of your study in a single paragraph and needs revisions.

Response:

Thank you very much for reviewing our manuscript and offering valuable comments on our manuscript. We have revised our manuscript following your comments and suggestions.

We removed last two paragraphs of the introduction and Figure 1 and kept in the study area section in line 86-108. We also created a last paragraph using your suggestion (general objective of this study) in line 79-83.

Best regards.
